# Manipulating dropout reveals an optimal balance of efficiency and robustness in biological and machine visual systems

**Jacob S. Prince** [1] [*], **Gabriel Fajardo** [2], **George A. Alvarez** [1] **& Talia Konkle** [1]
[1] Harvard University, Department of Psychology
[2] Boston College, Department of Psychology and Neuroscience

## Abstract

According to the efficient coding hypothesis, neural populations encode information optimally when representations are high-dimensional and uncorrelated. However, such codes may carry a cost in terms of generalization and robustness. Past empirical studies of early visual cortex (V1) in rodents have suggested that this tradeoff indeed constrains sensory representations. However, it remains unclear whether these insights generalize across the hierarchy of the human visual system, and particularly to object representations in high-level occipitotemporal cortex (OTC). To gain new empirical clarity, here we develop a family of object recognition models with parametrically varying dropout proportion ($p$), which induces systematically varying dimensionality of internal responses (while controlling all other inductive biases). We find that increasing dropout produces an increasingly smooth, low-dimensional representational space. Optimal robustness to lesioning is observed at around 70% dropout, after which both accuracy and robustness decline. Representational comparison to large-scale 7T fMRI data from occipitotemporal cortex in the Natural Scenes Dataset reveals that this optimal degree of dropout is also associated with maximal emergent neural predictivity. Finally, using new techniques for achieving denoised estimates of the eigenspectrum of human fMRI responses, we compare the rate of eigenspectrum decay between model and brain feature spaces. We observe that the match between model and brain representations is associated with a common balance between efficiency and robustness in the representational space. These results suggest that varying dropout may reveal an optimal point of balance between the efficiency of high-dimensional codes and the robustness of low dimensional codes in hierarchical vision systems.

## 1 Introduction

Biological and artificial visual systems must convert rich high-dimensional input data into a more compact, meaningful representational format to support downstream tasks. A core tension in this process relates to the nature of the sensory coding scheme. On one hand, the efficient coding hypothesis posits that visual systems should employ sparse, high-dimensional representations to capture the statistical regularities of the natural world with minimal redundancy (Barlow, 1959; Olshausen & Field, 1996; Simoncelli & Olshausen, 2001; Olshausen & Field, 2004; Ganguli & Sompolinsky, 2012). Alternatively, distributed coding schemes implement a broader distribution of information across neurons or computational units, thereby enhancing robustness and generalizability at the expense of efficiency (Shadlen & Newsome, 1998; Reich et al., 2001; Haxby et al., 2001; Cunningham & Yu, 2014). It remains unclear how the human visual system navigates this trade-off to function optimally in naturalistic contexts. To address this gap, here we directly instantiate this theoretical tension in a family of object recognition neural network models, which enables us to study where the human visual system falls within the sparse-to-distributed coding continuum.

**Sparse coding** schemes aim to maximize representational capacity by employing efficient information encoding with minimal redundancy (Barlow, 1959). Prior work suggests the benefit of sparse

---

[*] Corresponding author: jprince@g.harvard.edu

codes as a basis feature set in early stages of visual processing (Olshausen & Field, 1997; Vinje & Gallant, 2000) and for tasks like few-shot learning (Sorscher et al., 2022). While useful for representing the complexity of sensory inputs with fine-grained detail, such codes may be vulnerable to noise and perturbations, potentially yielding unstable population responses given small input changes (Stringer et al., 2019). Furthermore, extracting meaningful information from sparse representations may impose a significant readout challenge for downstream neurons or computational units. Precisely because the information is localized, readout depends on knowing "where to look" within the large, high-dimensional space to extract the relevant signals.

In contrast, **distributed coding** schemes involve a more expansive set of units in representing a given stimulus, thereby introducing redundancy into the population code (Reich et al., 2001). Distributed codes reside in lower-dimensional spaces due to the inherent redundancy among the neurons' tuning, which leads to correlated activity that can be captured with fewer unique dimensions (Cunningham & Yu, 2014). Empirical studies suggest that the visual hierarchy of primate inferotemporal cortex (IT) may pressure toward low-dimensional representations of object information (Lehky et al., 2014; DiCarlo & Cox, 2007). Others have theorized that lower-dimensional codes offer benefits of adversarial robustness (Amsaleg et al., 2020; Kong et al., 2022) and improved separability of object class manifolds (Cohen et al., 2020; Sorscher et al., 2022). While such coding strategies may afford these advantages, increased redundancy may eventually begin to impose a representational cost, constraining the system's capacity to encode a diverse range of important features.

To study these influences on sensory representation, related work has attempted to quantify the dimensionality of visual population responses within both biological and artificial systems. For instance, investigation of large-scale data from mouse primary visual cortex (V1) led to the proposal that particular dimensionalities, quantified as rates of eigenspectral decay, may reflect an optimal balance between high- and low-dimensional coding schemes (Stringer et al., 2019). This finding challenges the longstanding view that V1 implements maximally sparse codes. Instead, it suggests a more complex framework for understanding visual coding strategies, one that incorporates a balance between high- and low-dimensional codes. Despite these advances, substantial gaps remain in our understanding of coding principles across the human visual hierarchy.

Deep neural networks (DNNs) now offer a powerful framework to operationalize and scrutinize this sparse-to-distributed coding continuum **(Figure 1A)**. Here, we do so by developing a class of models where a single inductive bias—dropout regularization (Hinton et al., 2012)—is varied while keeping other inductive biases constant, isolating the impact of dropout on representation learning. Specifically, we train models with stochastic removal of neurons during each training iteration, varying the proportion of dropout across models **(Figure 1B)**. This regularization method is known to limit overfitting and enhance model generalization (Baldi & Sadowski, 2013; Srivastava et al., 2014; Liu et al., 2023).

The logic here is that, with increasing proportions of dropout, models will be pressured against assigning high weights to individual connections, since those units might not be consistently available. Thus, in object recognition models, information should consequently becomes more broadly distributed across the layer. As a result, increasing dropout $p$ should result in a smoother, lower-dimensional geometry of the population code. Our study first verifies these representational signatures empirically. Then, we leverage this form of controlled model variation to examine the effect of dropout on representational capacity, geometry, lesioning robustness, and emergent brain predictivity. We report three novel findings:

1. We confirm that "controlled model rearing" with increasing dropout $p$ yields models with increasingly distributed codes and lower-dimensional eigenspectra.

2. Varying dropout preserves task performance while systematically altering representational geometry, revealing an optimal point of robustness to simulated model "lesions".

3. Model-to-brain comparison with high-resolution fMRI data reveals that the most robust model also shows the highest degree of emergent brain alignment.

## 2 RESULTS

### 2.1 INCREASING DROPOUT REDUCES DIMENSIONALITY OF MODEL REPRESENTATIONS

To investigate the relationship between dropout regularization and the dimensionality of learned representations, we trained a family of models with controlled dropout regularization. The base encoder model was an AlexNet architecture (Krizhevsky et al., 2012) with dropout implemented at fully connected layers fc6 and fc7. We trained a family of 10 models, with dropout probabilities ($p$) varying from 0 to 0.9, incremented in steps of 0.1 **(Figure 1B)**.

All models were trained on the ImageNet dataset for 1000-way object classification (Deng et al., 2009), using the standard cross-entropy loss function. Models were trained using stochastic gradient descent on three Nvidia A100 GPUs with 40 GB RAM. Efficient data-loading was achieved via the Fast-Forward Computer Vision (FFCV) library (Leclerc et al., 2023). We used a one-cycle learning rate schedule, linearly ramping from an initial learning rate set to $1 \times 10^{-4}$ times the peak learning rate (0.05), which was reached at epoch 15, before linearly descending to $2 \times 10^{-5}$ over the remainder of training. Image resolution started at 160, then increased to 192 on epoch 71 (implemented as a linear ramp between epochs 65 and 76, rounded to the nearest multiple of 32). Given that dropout was already implemented as a regularization method, no further weight decay was applied. All training runs used a batch size of 512 and momentum value of 0.9.

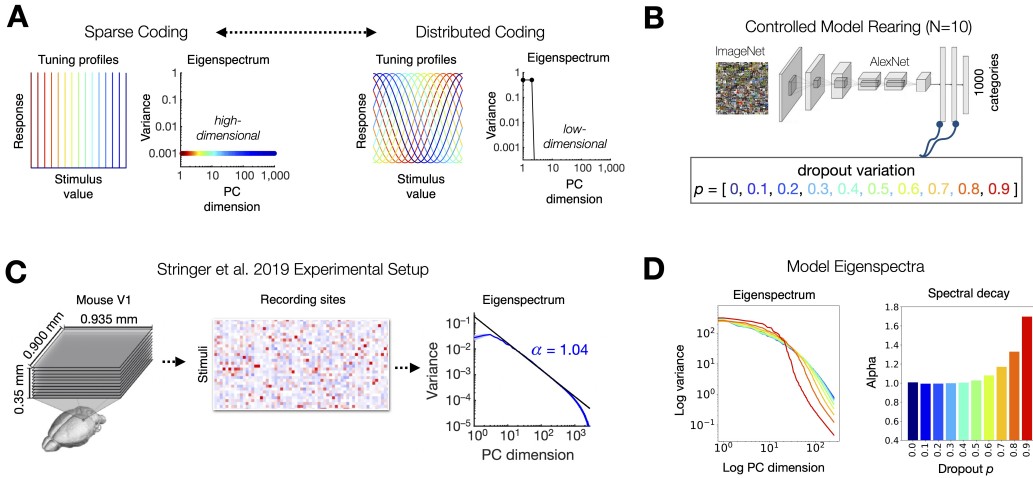

Figure 1: **Controlled model rearing with varied dropout $p$ yields variation between sparse and distributed coding schemes.** (A) Schematic depiction of the tension between sparse, high-dimensional codes and distributed, low-dimensional codes. Images adapted from Stringer et al., 2019. (B) Overview of controlled model rearing procedure. 10 models were trained with varying dropout $p$ from 0 to 0.9, in increments of 0.1. (C) Experimental approach from Stringer et al., 2019. Calcium imaging was used to record the simultaneous population activity of thousands of neurons in mouse primary visual cortex (V1) in response to natural images. From these brain responses, the neural eigenspectrum was estimated for early visual cortex. The alpha (rate of spectral decay) was estimated with a linear fit over the log eigenspectrum. (D) For each of the dropout models, the log eigenspectrum of layer fc6 activations (measured at inference time, with dropout disabled) is shown (left plot), with the first 250 dimensions plotted. The corresponding alpha values are shown for each model (right plot).

First we assessed the dimensionality of learned representations over the set of 10 models. To do so, we computed the activation matrices over a sample of 1000 images from the ImageNet validation set focusing on AlexNet layer fc6, which has been previously shown to have a high degree of brain alignment (Prince et al., 2023; Bao et al., 2020). From these activations, we followed a similar approach to that of Stringer et al., 2019, which quantifies the decay rate of the log-transformed eigenspectrum **(Figure 1C)**, as follows. Activation data were standardized to zero-mean and unit-variance. The PCA-derived eigenspectrum was then sorted in decreasing order, normalized by its

sum, and truncated to the first 250 components for subsequent analysis. A power-law fit to this eigenspectrum was computed as:

$$\log(\lambda_i) = \alpha \log(i) + \beta \tag{1}$$

Optimal parameters were estimated via weighted least squares regression, which aims to emphasize the importance of the leading principal components by inversely weighting them according to their rank, thereby ensuring a more robust estimation of the decay rate $\alpha$ in the eigenspectrum:

$$\mathbf{b} = (\mathbf{X}^\top \mathbf{W} \mathbf{X})^{-1} \mathbf{X}^\top \mathbf{W} \mathbf{y} \tag{2}$$

Here, $\mathbf{W}$ is a diagonal weight matrix with $w_i = \frac{1}{i}$, $\mathbf{y} = \log(\lambda_{\text{range}})$, and $\mathbf{X}$ contains $-\log(i)$ and a constant term. The parameter $\alpha$, which is the first element of the resulting $\mathbf{b}$, serves as our metric for the rate of eigenspectrum decay, indicating the effective dimensionality of the feature space.

The resulting eigenspectra for the parameterized set of dropout models are shown in **Figure 1D**. These show systematic variation in the resulting shape of the eigenspecta, with corresponding variation in the estimated alpha parameters. We observed a systematic reduction in dimensionality of object representations with increasing dropout $p$. These results confirm that dropout proportion serves as an effective knob for controlling the dimensionality of the learned representations.

## 2.2 VARYING DROPOUT ALTERS REPRESENTATIONAL GEOMETRY WHILE PRESERVING TASK PERFORMANCE

Next, we examined the impact of dropout variation on both task performance and on the nature of the internal representations learned across the layers of these models. Model performance on the ImageNet dataset, using the top-5 accuracy metric, is shown in **Figure 2A**. Despite the extensive range of dropout rates, all models exhibited surprisingly competent performance. Peak accuracy was observed for models trained with a dropout $p$ of 0.5. (We note that the original AlexNet model included dropout $p$=0.5, likely based on a hyperparameter sweep; our results converge with this setting). However, we also note that all models have performance within a 10% accuracy range. Thus, all models are capable of effectively learning the task at hand, despite a wide range of dropout probability.

### 2.2.1 REPRESENTATIONAL TRAJECTORY ANALYSIS

We next sought to understand the impact of dropout variation on the internal representations learned across the layers of these models. To do so, we used Representational Trajectory Analysis (Kallmayer et al., 2020, see also Mehrer et al., 2020). This method creates a visualization of the "representational trajectory" that different models take from the pixel space to their final representational format (**Figure 1B**). In this visualization, model layers with similar representational geometries are nearby in the plot, and model layers with more different geometries are further away.

To apply this analysis, for each layer $L$ of every model, activations were extracted to image set $I$, consisting of 250 sample images from the validation set. This yielded an activation matrix $\mathbf{A}_L$ where each row corresponds to the activation pattern elicited by a single image. Next, the Pearson dissimilarity for this layer was computed to form a representational dissimilarity matrix $\text{RDM}_L$, a square matrix that captures the dissimilarity between activation patterns for all image pairs:

$$\text{RDM}_L(i, j) = 1 - \frac{\text{cov}(\mathbf{A}_L(i, :), \mathbf{A}_L(j, :))}{\sigma(\mathbf{A}_L(i, :))\sigma(\mathbf{A}_L(j, :))} \tag{3}$$

Next, for each $\text{RDM}_L$, the vectorized lower triangular portions was extracted, and concatenated to construct the matrix $\mathbf{M}$. Finally, the Pearson dissimilarity between all pairs of rows in $\mathbf{M}$ was computed, yielding a Meta-Dissimilarity Matrix (MDM):

$$\text{MDM}(m, n) = 1 - \frac{\text{cov}(\mathbf{M}(m, :), \mathbf{M}(n, :))}{\sigma(\mathbf{M}(m, :))\sigma(\mathbf{M}(n, :))} \tag{4}$$

This MDM matrix reflects how dissimilar every model layer's representation is from every other model layer, across all models. To visualize the structure in this matrix intuitively, the MDM matrix

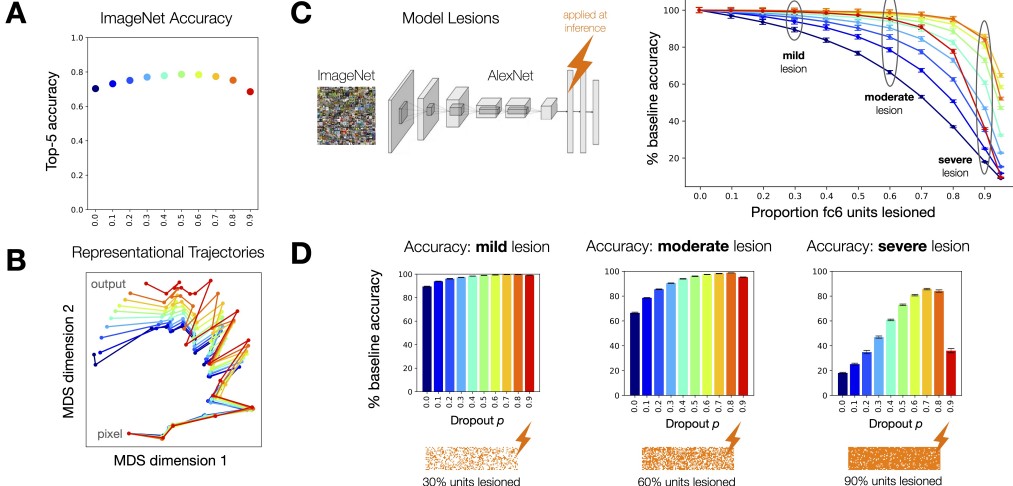

Figure 2: **Varying dropout preserves task performance while altering representational geometry, revealing an optimal point of robustness to simulated lesioning.** (A) Top-5 ImageNet accuracy after 100 training epochs, across the dropout models. (B) Representational Trajectory Analysis (Kallmayer et al., 2020): showing the layer-to-layer representational dissimilarity structure of all 10 models projected into 2D space using multidimensional scaling (MDS), with traces connecting the input and output stages of each model. (C) Model lesioning procedure. Percent of baseline top-5 ImageNet accuracy is plotted as a function of the proportion of units randomly lesioned in layer fc6. Error bars reflect the standard deviation accuracy over the 10 iterations. (D) Bar graphs summarize the observed outcomes for mild (20% layer units), moderate (50% layer units), and severe (80% layer units) lesions.

was input to 2D multidimensional scaling (MDS), yielding x-y coordinates for each model layer, which are shown in a 2D scatter plot. Layers from the same model are connected with a line, visualizing the 'representational trajectory' from the input (pixel space) to output (fc8). In this space, the farther apart two model layers are, the more different their representational geometries.

Inspection of these trajectories in **Figure 1B** reveals systematic variations in representational geometries learned across the layers of these models. The largest variation occurs predominantly in the fully connected layers, where dropout was explicitly applied. Interestingly, representational differences across these models also extended to the earlier convolutional layers, underscoring the cascading impact of localized dropout across the hierarchical structure of the network.

### 2.3 OPTIMAL ROBUSTNESS TO LESIONS IS OBSERVED AT HIGH (BUT NOT EXTREME) LEVELS OF DROPOUT

We next sought to understand the functional consequences of varying dropout. To do so, we applied simulated "lesions" to these models, drawing upon neuropsychological paradigms that assess the impact of damage to object-responsive cortex (Barton et al., 2002; Moro et al., 2008; Kanwisher & Barton, 2011). Note that dropout regularization occurs during training but not during inference. In this analysis, we are effectively applying dropout during inference time, as a way of lesioning or damaging the model. Presumably, the models trained with the most dropout will be the most robust to lesions during inference.

For this analysis, lesions were applied to layer fc6 in each model during inference (see Appendix A.3 for a similar analysis of fc7). Randomized groups of units were lesioned by setting their outputs to zero, where we varied the scope of the lesion between 0 to 95% of layer units, in steps of 10%. Top-5 ImageNet accuracy was subsequently re-evaluated. This lesioning protocol was performed iteratively 10 times, each with a different random subset of units targeted. This approach aligns with prior work examining the robustness of neural networks to selective perturbations (Cheney et al., 2017; Casper et al., 2021).

When lesioning the model trained with no dropout, a monotonic decline in performance was observed with increasing lesion sizes, as expected (**Figure 2C-D**). As the dropout rate increased, so did robustness, enabling substantial proportions of units in layer fc6 to be lesioned with minimal decrement in performance. This trend, however, reached an inflection point around $p = 0.7$, beyond which the models exhibited brittle behavior under large lesions. These findings validate that dropout can effectively mitigate the damaging effects of lesions, especially at mid-to-high levels of $p$. However, the results show that it is not the case that optimal robustness is observed at the most extreme levels of dropout. And, it is not the case that this lesion-robust model ($p = 0.7$) is the same as the most accurate model where no dropout is applied at inference ($p = 0.5$).

Intriguingly, this analysis suggests the existence of an ideal dropout level that effectively balances the merits of both low- and high-dimensional codes within the DNN model (i.e., robustness and representational capacity). These *in silico* findings further hint that the human visual system may navigate a similar trade-off between sparsity and robustness. To address this possibility, we next examined how these models relate to representations measured in a condition-rich fMRI study of human object-responsive cortex.

## 2.4 OPTIMAL ROBUSTNESS IS ASSOCIATED WITH MAXIMAL HUMAN BRAIN ALIGNMENT

The aim of our next analysis was to compare the emergent alignment between late-stage representational geometries of our dropout models and the representational geometry measured in human high-level visual cortex. To do so, we leveraged fMRI recordings from the Natural Scenes Dataset (NSD, **Figure 3A**, Allen et al., 2022). NSD contains measurements of over 70,000 unique stimuli from the Microsoft Common Objects in Context (COCO) dataset (Lin et al., 2014) at high resolution (7T field strength, 1.6-s TR, $1.8mm^3$ voxel size). Here, we focused on the brain responses to the set of 515 COCO stimuli that all 8 subjects viewed 3 times across scan sessions. All responses were estimated using a new, publicly available GLM toolbox (GLMsingle; Prince et al., 2022), which implements optimized denoising and regularization procedures to accurately measure changes in brain activity evoked by experimental stimuli.

We targeted our analysis to object-responsive cortex ("occipitotemporal cortex" in humans–OTC, which is homologous to "inferotemporal" IT cortex in non-human primates). The sector was defined following the procedures in Conwell et al., 2022. Specifically, within a broad visual system mask ("nsdgeneral" ROI), the selected voxels included those in the mid-to-high ventral and lateral ROIs ("streams" ROIs), supplemented by voxels from 11 category-selective ROIs with a $t$-contrast statistic $> 1$. The 8 NSD subjects yielded an average data dimensionality of 34,195 in their OTC sectors. Importantly, this sector encompasses high-level object-responsive cortex only, and excludes data from earlier visual regions (e.g. V1-V4). Thus, the scope of our analysis goes beyond replicating the work of Stringer et al., 2019, which was limited to area V1 in mice.

To perform model-brain comparison, we used classical Representational Similarity Analysis (RSA; Kriegeskorte et al., 2008), schematized in **Figure 3A-B**. First, the brain activations to 515 images were transformed into a representational dissimilarity matrix (RDM), as in equation (3). Note that the representational geometry of this brain sector was highly reliable (Conwell et al., 2022), making this brain region a strong mapping target for our dropout models. The same 515 images were shown to each model, and activations from the fc6 layer (dimensionality = 4,096 units) were recorded, and converted into representational dissimilarity matrices (RDMs) following the same procedure. Finally, the lower triangular values of the model and brain RDMs were correlated, with Pearson $r$ values plotted in **Figure 3C.** Note that this classical RSA procedure differs from more flexible, and more typical, model-to-brain linking procedures, which allow for feature re-weighting to best try to warp the model representational geometry to fit the brain geometry. Here, no re-weighting is allowed, revealing the degree of emergent alignment between model and brain geometries.

The results in **Figure 3C** show that across the models, all showed some degree of representational alignment with the brain. However, surprisingly, and in parallel with our earlier findings, layer fc6 of the model trained with $p = 0.7$ dropout was not only most robust to lesioning, but also showed the highest degree of alignment with human brain representations. This finding adds further plausibility to the hypothesis that dropout manipulation instantiates a brain-relevant tradeoff between efficiency and robustness. Moreover, the apparent synergy between model robustness and emergent

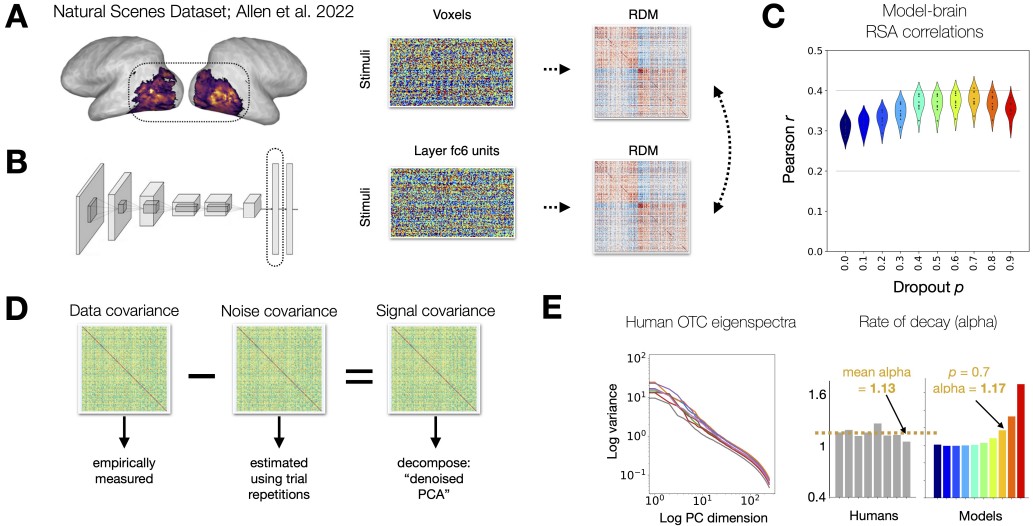

Figure 3: **Model-brain mapping reveals a link between lesion robustness and brain predictivity, associated with similar rates of spectral decay.** (A) Schematic overview of OTC data from the Natural Scenes Dataset (Allen et al., 2022). Highly reliable 7T fMRI responses to 515 Microsoft COCO images are subject to brain-model mapping via classical (unweighted) representational similarity analysis (cRSA). (B) Model layer fc6 features are extracted for the 515 test images and compared to brain RDMs using Pearson correlation. (C) Summary of model-brain cRSA prediction levels for the 8 NSD subjects, by dropout level. (D) Overview of the Generative Modeling of Signal and Noise approach (cvnlab, 2022). Signal covariance structure is estimated by subtracting empirically measured noise covariance (estimated using trial repetitions) from the full data covariance structure. Brain region eigenspectra are then computed directly from the signal covariance matrix. (E) Human OTC eigenspectra for the 8 NSD subjects (left), with alpha values summarized (gray bars, center) and compared to the dropout models (colored bars, right).

brain alignment raises an intriguing possibility: is this correspondence between neural systems also reflected in similar rates of eigenspectrum decay?

## 2.5 EIGENSPECTRUM OF OBJECT-RESPONSIVE CORTEX

To further probe the observed model-brain alignment, we next computed the rate of eigenspectrum decay of the brain representations, in order to compare to the model eigenspectra. Critically, while the models are noiseless systems, the brain measures are not. Thus, simply calculating the brain's eigenspectrum directly would also reflect the noise structure, which poses a thorny methodological challenge. For instance, if the noise present in the data were to exhibit a high-dimensional structure relative to the signal, then it could systematically bias naive eigenspectrum $\alpha$ estimates (e.g. computed by applying PCA directly to the data) in the downward direction. To address this issue, we employed a novel technique known as Generative Modeling of Signal and Noise (GSN, cvnlab, 2022). This approach leverages available stimulus repetitions to distinguish between signal and noise covariance structure, thereby yielding a more reliable, "denoised PCA" from which the brain's eigenspectra can be estimated.

### 2.5.1 GENERATIVE MODELING OF SIGNAL AND NOISE

For a given ROI, the GSN method estimates multivariate Gaussian distributions characterizing the signal and the noise under the assumption that observed responses can be characterized as sums of samples from the signal and noise distributions. This approach is similar in spirit to the cross-validated principal component analysis (cvPCA) method developed by Stringer et al., 2019, in that both techniques aim to separate signal-related variance from trial-to-trial noise. However, cvPCA achieves this by computing the covariance between training and test presentations of an identical

stimulus ensemble, to isolate stimulus-related variance. In contrast, GSN utilizes stimulus repetitions to model both the signal and noise explicitly, which aims to provide a more comprehensive description of the empirical data. As such, GSN provides a principled, unbiased method for extracting eigenspectra from noisy data.

To briefly summarize the approach: GSN assumes that for each condition, trial-to-trial variability (noise) adheres to a zero-mean multivariate Gaussian distribution, a property shared across conditions. Signal variability across conditions is also modeled as a multivariate Gaussian whose parameters are expected to vary from the noise distribution. The model can be summarized as:

$$
\begin{aligned}
D &\sim X_1 + X_2, \\
X_1 &\sim \mathcal{N}(\mu_{\text{signal}}, \Sigma_{\text{signal}}), \\
X_2 &\sim \mathcal{N}(0, \Sigma_{\text{noise}}).
\end{aligned}
\tag{5}
$$

Here, $D$ denotes the observed multivariate response for a given trial, $X_1$ and $X_2$ represent the multivariate responses due to signal and noise, respectively. Parameters $\mu_{\text{signal}}$ and $\Sigma_{\text{signal}}$ specify the mean and covariance of the signal Gaussian, whereas $\Sigma_{\text{noise}}$ is the covariance of the noise Gaussian. This modeling framework is generative, as it directly characterizes the data-generating processes.

The crux of the GSN approach lies in parameter estimation for both signal and noise distributions, to be fully described and validated in a forthcoming manuscript. Here we provide a brief, non-exhaustive overview of the GSN algorithm, which capitalizes on the property that the sum of samples from two Gaussian distributions is also Gaussian-distributed. First, to estimate the *noise* covariance, we start with a data matrix $X$ ($n$ fMRI voxels $\times c$ conditions $\times t$ trials). To estimate the noise structure, we calculate the covariance of voxel responses over stimulus repetitions separately for each condition, and then average the covariance across conditions:

$$
\hat{\Sigma}_{\text{noise}} = \frac{1}{c} \sum_{j=1}^{c} \text{cov}(X_j)
\tag{6}
$$

Next, we estimate the *data* covariance of voxel responses after averaging over stimulus repetitions:

$$
\hat{\Sigma}_{\text{data}[n]} = \text{cov}(\bar{X})
\tag{7}
$$

Finally, to infer the *signal* covariance, we subtract the estimated noise covariance from the estimated data covariance (**Figure 3D**). Due to averaging across $t$ trials, the variance of the noise is reduced by a factor of $t$ in the trial-averaged data:

$$
\hat{\Sigma}_{\text{signal}} = \hat{\Sigma}_{\text{data}[n]} - \frac{\hat{\Sigma}_{\text{noise}}}{t}
\tag{8}
$$

The GSN procedure next implements a check for whether the estimated covariance matrices are valid and if not, computes the nearest (Frobenius matrix norm) symmetric positive semi-definite matrix to a given square matrix (Higham, 1988; see cvnlab, 2022).

Once the signal covariance structure, $\Sigma_{\text{signal}}$, was estimated for OTC data from each NSD subject, we directly computed its eigenspectrum using matrix decomposition. **Figure 3E** shows the eigenspectrum of object-responsive cortex seperately for each of the 8 participants. As was previously observed in mouse V1 (Stringer et al., 2019), we see a characteristic linear fall-off on the log-log plot of the eigenvalues. The $\alpha$ decay was computed for each participant following the same procedure applied to the dropout models' eigenspectra, with a mean $\alpha = 1.13$ across participants (standard deviation $\alpha = 0.05$).

For comparison to the dropout models, we related these estimated $\alpha$ values to those arising from model layer fc6 across the model family (**Figure 3E**, see also Appendix sec. A.2). This analysis revealed that the model trained with $p = 0.7$ dropout, which was most robust to lesions and brain predictive, also demonstrated a rate of eigenspectrum decay ($\alpha$ value) that most closely matched that of human OTC. This observation adds a further layer of plausibility to the hypothesis that the advantages of dropout in model robustness and efficiency are not just computationally useful but also neurobiologically relevant. One implication of these results is that the brain's representation may not be overly dependent on single units or specific read-out connections. Further, the close match in $\alpha$ values suggests that the alignment between model and brain representations may be driven by similar constraints on efficiency and robustness within their respective feature spaces.

# 3   LIMITATIONS

Here we focused on dropout regularization as a way to operationalize a set of representations that vary along the sparse-to-distributed axis. However, it is possible that the percentage of dropout may reflect a pressure to be more-or-less distributed (rather than pushing towards sparsity directly). Alternative regularization techniques such as L1 and L2 penalties on the weights could also be explored. It is an open empirical question whether these regularization techniques have distinct consequences on the geometry of the representations, and their emergent robustness and brain correspondence.

Additionally, here we focused on the dimensionality and structure of the representation of block 6 of an AlexNet architecture, and object-responsive cortex within the human visual system. Given the correspondences in the emergent robustness, alignment, and eigenspectra, it will be informative to understand whether these spectral signatures of the representational space predict emergent brain alignment across other model architectures, and across other stages of the visual hierarchy (Canatar et al., 2023; Conwell et al., 2022; Elmoznino & Bonner, 2022; Sorscher et al., 2022, Appendix A.4).

# 4   CONCLUSION

In this work, we have instantiated the theoretical tension between sparse and distributed coding schemes with controlled regularization over a set of deep neural network (DNN) models trained on object recognition. According to the efficient coding hypothesis, sparse, high-dimensional representations offer optimal information encoding at the expense of robustness (Olshausen & Field, 2004). Conversely, more distributed (lower-dimensional) coding schemes provide enhanced robustness by introducing representational redundancy, at some cost to information capacity (Stringer et al., 2019; Kong et al., 2022). Our approach involved "controlled rearing" of a set of models with varying dropout proportion ($p$) during training. Three novel findings emerged: (1) dropout variation systematically altered the dimensionality of model representations, operationalizing a continuum of more sparse to more distributed codes; (2) varying dropout influenced the learned representational geometries of these models, while generally preserving task performance, revealing an optimal point of robustness during simulated lesions at inference; and, (3) the model with optimal robustness to lesions was also associated with maximal emergent brain alignment, and comparable rates of spectral decay. Overall, these results suggest that varying dropout reveals an optimal point of balance between the efficiency of high-dimensional codes and the robustness of low dimensional codes in biological and artificial vision systems.

It is an open question what the parallel is between dropout regularization during model training and underlying neurobiological mechanisms. On one hand, dropout has been likened to synaptic failures, and variable neurotransmitter release probabilities in neural systems (Shadlen & Newsome, 1998; McKee et al., 2021). This is a rather direct mechanistic parallel (though, if so, this factor should be operating both at training and inference). However, another possibility is that the brain arrives at this representational format and dimensionality though completely different mechanisms, e.g. by interfacing with associative memory systems, which could pressure both for redundancy to aid generalization, and, for sufficient sparsity to avoid memory "collisions" during readout (Olshausen & Field, 2004; Palm, 2013). On this account, dropout mechanisms in models may have no relationship to the underlying biological operations, and still help promote brain-like formats by proxy, through their regularization of the learning process. Understanding how and why neurobiological systems may implement such forms of regularization is an important area for further investigation.

Broadly, the current work expands a growing body of research on the spectral properties of representations in artificial and biological visual systems. For instance, Canatar et al., 2023 have recently proposed a method for analytically relating the brain predictivity of a DNN model to its spectral characteristics, and to the orientation of its eigenvectors. In related work, Khosla et al., 2023 have introduced a framework for identifying "privileged axes" in brain and DNN representations, suggesting that such axes function to promote increased sparsity and efficiency of the neural code. And, Sorscher et al., 2022 have identified striking mismatches between the geometry and dimensionality of manifolds in the primate visual pathway and of trained DNNs, in the context of few-shot learning paradigms. Collectively, these studies reflect necessary progress beyond the binary debate of sparse versus distributed codes, and toward more nuanced theories of information coding in sensory systems, which can be tested empirically through controlled "rearing" of neural network models.

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

## ACKNOWLEDGMENTS

We thank all members of Harvard Vision Lab for helpful conversations throughout this project and for feedback on the manuscript. This work was funded by: Natural Science Foundation CAREER Grant #1942438 (TK); National Defense Science and Engineering Graduate Fellowship (JSP).

## A APPENDIX

### A.1 QUANTIFYING ACTIVATION SPARSITY

We examine both the lifetime and population sparsity of layer fc6 activations across the dropout models. Lifetime sparsity refers to the sparsity observed in a single neuron's activity over multiple inputs. The lifetime sparsity for a unit $u$ over a dataset of images $I$ is calculated as follows:

$$\text{Lifetime Sparsity}_u = \frac{1}{|I|} \sum_{i \in I} \chi_{\{x_{u,i}=0\}} \tag{9}$$

where $|I|$ represents the number of images, $x_{u,i}$ denotes the activation of unit $u$ for image $i$, and $\chi_{\{x_{u,i}=0\}}$ is the indicator function that is 1 if the unit is inactive (the activation is less than or equal to 0), and 0 otherwise.

Population sparsity describes the proportion of inactive neurons across the layer's response pattern for a single input. The population sparsity for an image $i$ over the set of units $U$ is given by:

$$\text{Population Sparsity}_i = \frac{1}{|U|} \sum_{u \in U} \chi_{\{x_{u,i}=0\}} \tag{10}$$

where $|U|$ is the total number of units, and $x_{u,i}$ is defined as before.

These metrics are shown in **Figure 4**, revealing that both lifetime and population sparsity decrease as dropout $p$ increases from 0 to 0.9. These results are consistent with the observation that the effective dimensionality of layer activations tends to decreases as dropout $p$ increases.

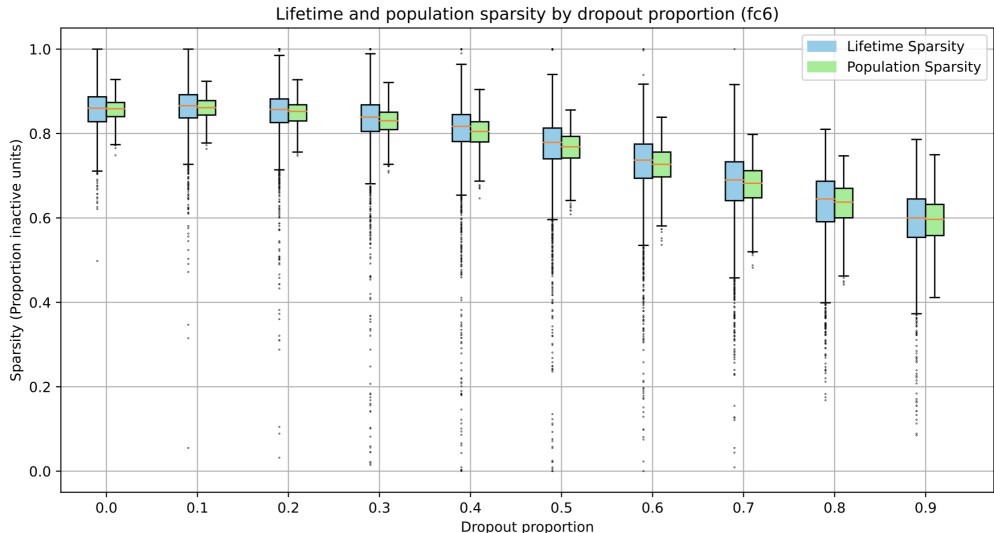

Figure 4: **Analysis of sparsity.** Activation sparsity levels in fc6 are reported for both lifetime and population sparsity metrics, plotted for models with increasing dropout along the x-axis. Each dot is an images, from the set of 1000 probe images (1 from each category).

## A.2 EIGENSPECTRA COMPARISON WITH DIFFERENT PROBE IMAGE SETS

In the main text, human OTC eigenspectra were assessed with 515 COCO images from the Natural Scenes Dataset, while the model eigenspectra was assessed with a sample of the validation set of ImageNet (in-distribution). Here we also assessed whether the model eigenspectra would be dependent on the choice of probe set. We directly compare fc6 eigenspectra computed over the ImageNet validation stimuli with those computed over the 515 COCO stimuli from the fMRI analysis. The results are shown in **Figure 5**, and show very similar outcomes across these two image datasets (average difference between $\alpha$ across layers = 0.02).

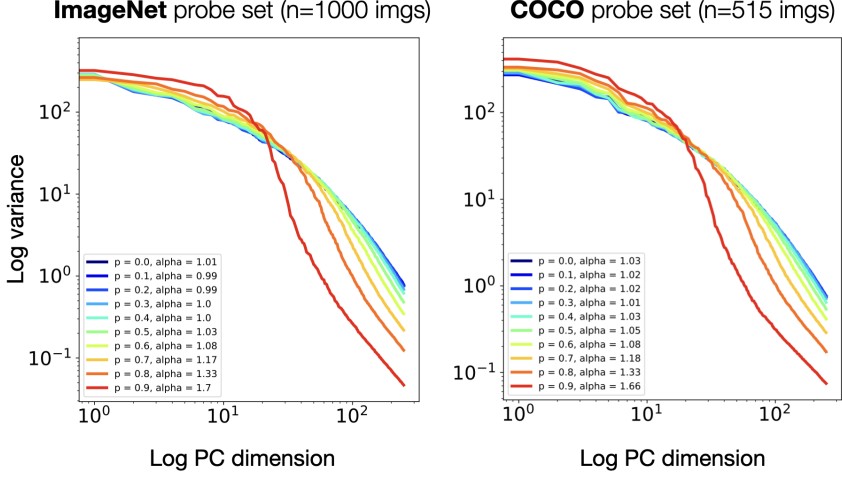

Figure 5: **Comparing ImageNet and COCO probe stimuli.** Layer fc6 eigenspectra and corresponding alpha estimates are plotted for model responses to a probe set of 1000 ImageNet validation stimuli (left) and a probe set of the 515 Microsoft COCO stimuli that were used for the fMRI analyses (right).

### A.3 ANALYSES OF LAYER FC7

A parallel set of post-hoc were analyses conducted on layer fc7, are shown in **Figure 6**. We find the model with $p = 0.7$ dropout again had the highest emergent brain alignment.

However, we note that this layer was generally less strongly correlated with the brain sector than layer fc6. Further, layer fc7 also showed less convergence across other signatures. For example, the eigenspectra of the fc7 layer of this model was generally steeper, meaning that the fc7 layer of a model with less drop-out best matched the slope of OTC eigenspectra ($p = 0.3$). Further, lesions to layer fc7 at inferences did not show a u-shape curve, and instead showed more robustness with increasing dropout proportion.

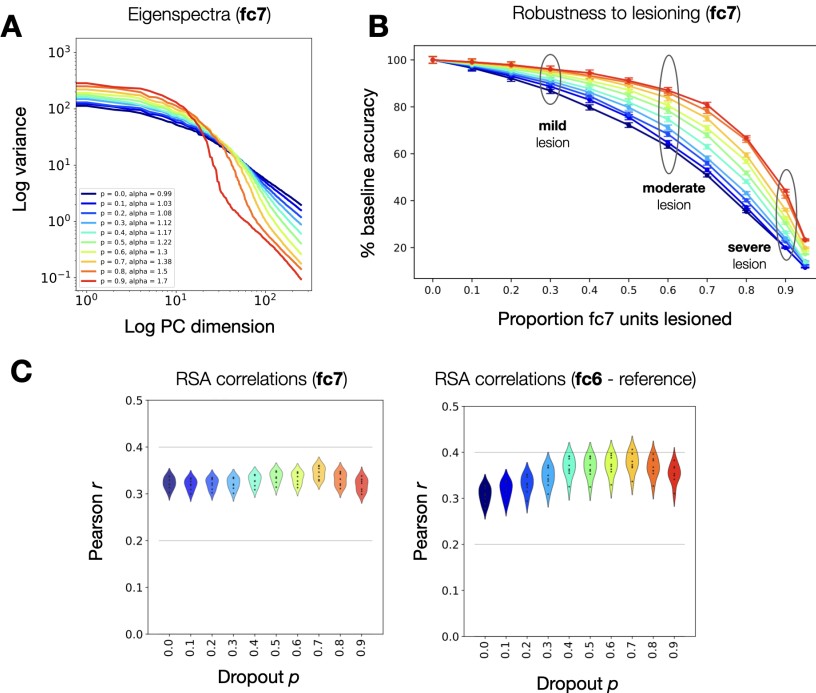

Figure 6: **Analysis of Layer fc7.** (A) fc7 eigenspectra are plotted with corresponding alpha estimates over the same probe set of 1000 ImageNet validation stimuli used for previous analyses. (B) Lesioning outcomes. Percent of each model's baseline top-5 ImageNet accuracy is expressed as a function of proportion fc7 units randomly lesioned at inference. (C) RSA correlations for layer fc7 are shown, with dots reflecting the 8 NSD subjects. (D) As a point of reference, RSA correlations for layer fc6 are shown.

There are several implications of these results. First, the features in fc7 are less well aligned to the OTC brain sector data than fc6. Second, these results also demonstrate that a simple measure of effective dimensionality (the slope of the eigenspectrum) is not itself a sufficient metric to predict emergent brain alignment. Further, it seems that lesions closer to the output stage show a more direct relationship between dropout proportion during training and lesioning during inference. Together these results highlight that eigenspectra slope, lesion robustness, and emergent brain alignment do not necessarily all hang together across all layers, providing room for future work to gain a deeper theoretical understanding of the relationship among these constructs (Canatar et al., 2023).

## A.4 REPLICATING ANALYSES USING HUMAN V1 DATA

We perform initial analyses to extend our paradigm from occipitotemporal cortex (OTC) to human V1. Specifically, we analyze data from the ventral aspect of V1 across the 8 NSD subjects (mean dimensionality = 2,650). Data processing and model-brain comparison occur in the same manner as described for comparisons with the OTC sector.

First, layer-wise analyses reveal that the 'maxpool1' stage of the dropout models is consistently the most predictive of human V1 activity (**Figure 7A**). Notably, the overall alignment of these models with V1 is lower compared to the observed fc6-OTC prediction levels. We also observe that varying amounts of dropout applied at the fc6 and fc7 layers does not significantly alter the representational characteristics in 'maxpool1', as shown in both representational trajectory analysis (**Figure 2B**) and in the effective dimensionality and emergent brain alignment assessments (**Figures 7B** and **7C**, respectively).

These findings suggest that manipulating dropout at earlier stages in the network, possibly within the first block, might be required in order to extend the current paradigm to study human V1 representations. Our initial findings indicate that V1 representations may be of lower dimensionality than those in the OTC (**Figure 7D**), providing a starting point for further investigation into the eigenspectra across different areas of the human visual system.

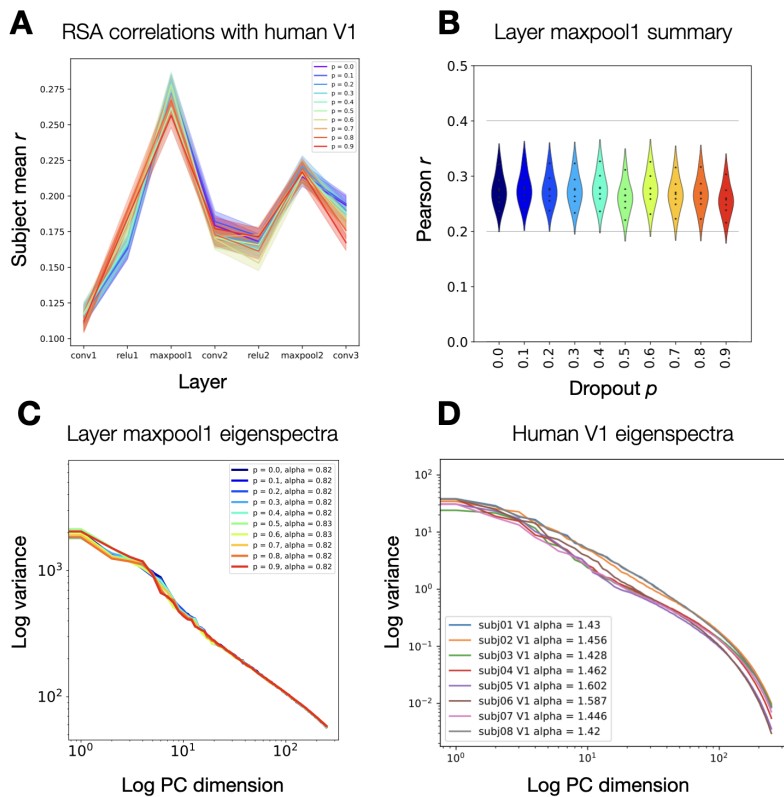

Figure 7: **Analysis of human V1 data.** (A) RSA correlations are summarized over the 8 NSD subjects for layers conv1 through conv3. Lines reflect the mean over subjects, with shaded regions reflecting the SEM. The same 515 test stimuli are used as in the primary analyses. (B) Complete RSA correlations for layer maxpool1 are shown, with dots reflecting the 8 subjects. (C) Model layer maxpool1 eigenspectra are plotted in log-log space, with alpha values reflecting the slope of the linear fit to the log eigenspectrum. Eigenspectra are computed over a probe set of 1000 ImageNet validation stimuli. (D) Human V1 eigenspectra estimated using the GSN technique. All analysis procedures are identical to those previously reported for OTC.

