# OpenReview forum: "Manipulating dropout reveals an optimal balance of efficiency and robustness in biological and machine visual systems"
_ICLR.cc/2024/Conference — ICLR 2024 poster_

### Official Review · Reviewer_YU1u · 2023-10-29

**Soundness:** 3 good
**Presentation:** 4 excellent
**Contribution:** 3 good
**Rating:** 6
**Confidence:** 4

**Summary:**

In this study, the researchers manipulated the degree of dropout in AlexNet, and then computed the eigenspectrum of the population responses in fc6, following Stringer, 2019. The results showed that increasing the degree of dropout reduced the representational dimension in fc6 and that dropout=0.7 achieved the best test results during the inference process. The authors also computed the eigenspectra of voxel responses in human visual cortex using NSD data and found that it was similar to AlexNet at dropout=0.7, while the correspondence between fc6 and voxel responses was strongest at dropout=0.7.

I am generally positive about this paper. It replicates the analysis of mouse V1 data in human fMRI data.

**Strengths:**

1. Using a large-scale fMRI dataset and computational analyses to address the coding principles of human visual cortex
2. A novel approach is developed to estimate the signal correlation matrix and and the noise correlation matrix in population responses

**Weaknesses:**

see my questions below

**Questions:**

1. In Figure 1, I am not sure why dropout was only implemented in fc6 and fc7 in model training? In a typical training, dropout was applied to all layers
2. I am wondering about the results of fc7 using the similar analysis in Figs. 2&3
3. In Figure 2, dropout=0.5 is the best in terms of top-5 ImageNet accuracy (Fig. 2A) but dropout=0.7 is the best in terms of accuracy with unit lession (Fig. 2D). It is debatable what exactly the criteria we emphasize when talking about coding efficiency.
4. In Figure 3, how many voxels and how many fc6 units are used? I suspect that the numbers are very different.
5. The original study by Stringer, 2019 only recorded neurons in mouse V1. However, the ROI used here included several low- and high-level cortex. Do you expect the eigenspectrum to be different across visual areas in humans. Actually, I would say that this is the novel point to make compared to simply replicating the analyses in Stringer, 2019.

---

> ### Author Response · Authors · 2023-11-21
>
> # Strengths:
> > *I am generally positive about this paper . . . It replicates the analysis of mouse V1 data in human fMRI data . . . Using a large-scale fMRI dataset and computational analyses to address the coding principles of human visual cortex. A novel approach is developed to estimate the signal correlation matrix and and the noise correlation matrix in population responses.*
>
> Thanks for the positive feedback!
>
> # Weaknesses / Questions:
> > *In Figure 1, I am not sure why dropout was only implemented in fc6 and fc7 in model training? In a typical training, dropout was applied to all layers*
>
> In the PyTorch implementation of AlexNet, dropout is applied only in layers fc6 and fc7 by default. As such, we used this default architecture for all experiments reported here. For future publication, we have begun a more systematic exploration of dropout applied to different groups of layers (or across the entire model), and the corresponding impact on model representations and alignment to biological visual systems.
>
> >*I am wondering about the results of fc7 using the similar analysis in Figs. 2&3*
>
> We have now added new analyses (Appendix A.3) showing the results of our main analyses using layer fc7 rather than fc6.
>
> > *In Figure 2, dropout=0.5 is the best in terms of top-5 ImageNet accuracy (Fig. 2A) but dropout=0.7 is the best in terms of accuracy with unit lesion (Fig. 2D). It is debatable what exactly the criteria we emphasize when talking about coding efficiency.*
>
> Good observation. Here, our specific goal is to identify where human visual representations fall along the continuum of dropout models, rather than finding the overall best ImageNet model. As such, we highlight the p=0.7 model, which stands out both in its peak brain alignment and its robustness to unit failure (this is relevant since neurobiological circuits are known to be noisy/stochastic). But, in Section 2.3, we now address this trade-off between accuracy and robustness.
>
> > *In Figure 3, how many voxels and how many fc6 units are used? I suspect that the numbers are very different.*
>
> The brain data from the 8 NSD subjects has an average dimensionality of 34,195 within the large-scale occipitotemporal cortex (OTC) ROI, and layer fc6 of AlexNet has 4,096 units. While these values do differ by an order of magnitude, our RSA procedure is quite standard for comparing brain sector and DNN layer representations. We have added these quantifications of voxel and unit counts to Section 2.4 of the paper.
>
> > *The original study by Stringer, 2019 only recorded neurons in mouse V1. However, the ROI used here included several low- and high-level cortex. Do you expect the eigenspectrum to be different across visual areas in humans. Actually, I would say that this is the novel point to make compared to simply replicating the analyses in Stringer, 2019.*
>
> First, we now more clearly note that the human OTC ROI encompasses high-level visual cortex only (Section 2.4). Lower-level visual regions such as V1-V4 are not included.  We now emphasize the novelty of our results, which are not simply a replication of Stringer et al., 2019.
>
> Second, we have now taken preliminary steps to replicate our analysis of OTC using human V1 data. Indeed, the average rate of the human V1 eigenspectrum decay (alpha) is 1.48, while the average alpha for human OTC is 1.13. We now detail several new analyses on human V1 in a new Appendix 4. This provides an initial clue that the reviewer’s suggestion of studying eigenspectra in different areas may indeed yield productive insights. We are now working on training models with varied dropout at earlier stages (e.g. in the first block) to further meaningfully extend our paradigm to human V1.
>
> # Summary:
> We have added new analyses and provided detailed explanations to address your questions and concerns. We’ve also extended our study to include preliminary analysis of human V1 data, in an attempt to assess the generality of our paradigm. We appreciate your insightful comments and hope these revisions encourage you to raise your score.

---

### Official Review · Reviewer_h1kR · 2023-11-01

**Soundness:** 3 good
**Presentation:** 4 excellent
**Contribution:** 3 good
**Rating:** 6
**Confidence:** 4

**Summary:**

To understand the trade-off between high-dimensional and uncorrelated efficient codes and distributed codes known better for generalization and robustness, the authors trained a family of object recognition models with parametrically varying dropout proportions. They found that a higher proportion of dropouts results in more smooth and low-dimensional representational space, with 70% of dropouts offering optimal robustness (against simulated brain lesions). Interestingly, this is also associated with the highest degree of emergent brain alignment for fMRI data in humans. Furthermore, the match between the model and brain representations is associated with a common balance between the efficiency of a high-dimensional code and the robustness of a low-dimensional code.

**Strengths:**

The idea and approach are simple, and the results are compelling and conceptually reasonable. The study illuminates the relationship between sparse codes and distributed codes. Using simulated lesions to evaluate robustness is an interesting innovation. The paper is well written. The presentation is clear and well-organized. The best alignment between human fMRI representation and of the representation of the model with the greatest robustness is very interesting, suggesting robustness, in addition to efficiency, is indeed a very important criterion for learning brain representation. So, the study does provide insights into the brain. I rank this an acceptable paper, with "grade" somewhere between 6 and 7, so I round it up to 6, because there is no "7".

**Weaknesses:**

The idea seems obvious and intuitive, and the approach and the work are perhaps too simple.
The work is most empirical and does not have much theoretical analysis.

**Questions:**

While it is still fantastic to prove empirically a simple and intuitive idea is true elegantly,  is it possible to show this analytically?  What would be the analytical approach?

---

> ### Author Response · Authors · 2023-11-21
>
> # Strengths:
> > *... the results are compelling and conceptually reasonable. … Using simulated lesions to evaluate robustness is an interesting innovation. The paper is well written. The presentation is clear and well-organized … the study does provide insights into the brain. I rank this an acceptable paper, with "grade" somewhere between 6 and 7, so I round it up to 6, because there is no "7".*
>
> Thank you!
>
> # Weaknesses & Questions:
>
> > *The idea seems obvious and intuitive, and the approach and the work are perhaps too simple.
> The work is most empirical and does not have much theoretical analysis. … While it is still fantastic to prove empirically a simple and intuitive idea is true elegantly, is it possible to show this analytically? What would be the analytical approach?*
>
> True, the work is all empirical, focusing on insights gained through comparing a controlled set of parameterized dropout models. Given the simplicity of the idea and the promising, novel emergent correspondances with brain responses, we hope this work prompts others to dig into the analytical underpinnings of the results here.

---

### Official Review · Reviewer_yavH · 2023-11-01

**Soundness:** 4 excellent
**Presentation:** 3 good
**Contribution:** 3 good
**Rating:** 6
**Confidence:** 3

**Summary:**

The paper introduces a neurobiological link between dropout, a common ML regularization method and human brain processing.

Key findings include:
* There was a reduction in object representation dimensionality with increased dropout levels for NNs
* NN performance was relatively consistent across different levels of dropout, first monotonically increasing and then monotonically decreasing
* Representational trajectory analysis revealed most differences in the fully connected layers where dropout was applied
* Dropout mitigates effect of [neuron] lesioning, but to a limited extent
* A model that was most robust to [neuron] lesioning (dropout p=0.7) mostly closely matched human brain representations obtained from an FMRI study and evaluated using Representational Similarity Analysis (RSA).

**Strengths:**

The paper introduces a systematic analysis of dropout and [neuron] lesioning with a neuro-biological perspective in mind. Experiments highlight interesting and novel findings with respect to model variations due to dropout and newly discovered connections to human brain processing. Overall, the paper makes advances in understanding spectral properties learnt by humans and neural networks.

Authors explain the limitations of their work in terms of considering only one type of regularizations.

**Weaknesses:**

The paper would benefit from more clarity and better organization. It was somewhat difficult to understand key components of the paper. For instance, the definition of “lesioning”, i.e., pruning neurons in an NN layer, which is central to the paper message, is found in a figure caption (Figure 2), but should appear in the main paper.

Specific points:
* It is not clear how the layer for lesioning (relu6) was selected or whether results will still hold if a different layer would be selected.
* Results are reported using a single architecture (AlexNet), and may not generalize to other NNs. Also, the statement in the beginning of conclusion “schemes within a family of deep neural network (DNN)” needs to be adapted to better reflect the type of networks considered. Specifically, here family seem to refer not to architecture variations, but changes in the level of dropout within an architecture.
* Authors should discuss the significance of the reported findings for the ML community. I thought that many of the reported analyses were cleverly executed, but I struggled to understand the main message behind the paper.

**Questions:**

I did not understand why human responses were compared to network responses but using different datasets (as discussed in section 2.4, human fMRI was obtained using Natural Scenes Dataset and compared to results on ImageNet using networks)? Authors should either explain the motivation of not running networks on the Natural Scenes dataset to match human fMRI, or report results using a single dataset.

It would be useful to understand the effect of dropout on the trajectories from representational trajectory analysis reported in Figure 1B. Specifically, is it dropout that induces the highest variation in the fc layers, or the underlying network architecture?

---

> ### Author Response · Authors · 2023-11-21
>
> # Strengths:
> > *The paper introduces a systematic analysis of dropout and [neuron] lesioning … Experiments highlight interesting and novel findings … with newly discovered connections to human brain processing. Overall, the paper makes advances in understanding spectral properties learnt by humans and neural networks.*
>
> Thanks for the positive comments!
>
> # Weaknesses:
> > *For instance, the definition of “lesioning”, i.e., pruning neurons in an NN layer, which is central to the paper message, is found in a figure caption (Figure 2), but should appear in the main paper*
>
> We have clarified this method in the main text in the revised manuscript. Thank you.
>
> > *It is not clear how the layer for lesioning (relu6) was selected or whether results will still hold if a different layer would be selected.*
>
> True. We now clarify that we focused on lesioning fc6 during inference as a first probe, and have now added a new Appendix section A.3 with the results for lesioning layer fc7. (We note that lesioning fc6 and relu6 have the same functional consequence in our implementation, so we now refer to the fc layers rather than the relu layers for our lesioning results.)
>
> > *Results are reported using a single architecture (AlexNet), and may not generalize to other NNs.*
>
> True. We raise this in the limitations sections.
>
> > *Also, the statement in the beginning of conclusion “schemes within a family of deep neural network (DNN)” needs to be adapted to better reflect the type of networks considered. Specifically, here family seem to refer not to architecture variations, but changes in the level of dropout within an architecture.*
>
> We have attempted to clarify this sentence: "In this work, we have instantiated the theoretical tension between sparse and distributed coding schemes with controlled regularization over a set of deep neural network (DNN) models trained on object recognition."
>
> # Questions:
> > *I did not understand why human responses were compared to network responses but using different datasets (as discussed in section 2.4, human fMRI was obtained using Natural Scenes Dataset and compared to results on ImageNet using networks)? Authors should either explain the motivation of not running networks on the Natural Scenes dataset to match human fMRI, or report results using a single dataset.*
>
> True. We thought characterizing the eigenspectra of the model layer with in-distribution images made sense but we very much see your point. So, we now also report a new analysis using the same 515 COCO images from the fMRI study to compute and compare the eigenspectra in the model layer and brain data.  The ImageNet and COCO probe sets yield nearly identical results (mean difference in alpha value of 0.02 over the 10 models’ fc6 layers).  These results are now reported in Appendix A.2.
>
> > *It would be useful to understand the effect of dropout on the trajectories from representational trajectory analysis reported in Figure 1B. Specifically, is it dropout that induces the highest variation in the fc layers, or the underlying network architecture?*
>
> The network architectures are all the same across these plots, so the variation in their representational trajectories is due to differences in dropout rates during training.
>
> # Summary:
> Thanks for your thoughtful feedback. We have attempted to address as many points as possible given the time constraints of the response period.

---

### Official Review · Reviewer_wjr3 · 2023-11-11

**Soundness:** 3 good
**Presentation:** 3 good
**Contribution:** 3 good
**Rating:** 6
**Confidence:** 3

**Summary:**

The paper use dropout as an approach to control the sparsity of neural representation in deep neural network (AlexNet), and evaluated the robustness of the resulting networks to lesions to one of the layers. The dropout rate of 0.7 was found to be most robust to lesion at inference time. Coincidently, at this level of dropout, the geometry of neural representation in the same layer also shows the highest correlation to the neural representational geometry to a set of COCO dataset images, in occipitaltemporal cortex of human. Further investigating the rate of decay in the eigenspectrum of the brain region and those of DNNs with different dropout rates, the decay rate of the model with 0.7 dropout rate also matches the decay rate of the human brain. Together, the result appears to suggest that, if dropout rate can be considered as a valid way of controlling for the sparsity of neural encoding, then perhaps the brain chooses its level of sparsity at a level most robust to lesioning of "neurons".

**Strengths:**

The result shows a high consistency among three results: the dropout rate yielding the highest robustness of the DNN against high-level lesion, the dropout that yields neural representation best matching that of the object-recognition related brain region, and the the dropout rate yielding the same decay rate in the eigenspectrum of the covariance structure of fMRI spatial patterns across images.

It used a novel approach to separate the spatial covariance structure due to potentially noise and that due to signal in fMRI activity

**Weaknesses:**

Although the decay rate of eigenspectrum is an intuitive way of showing the effective dimensions being used to encode stimuli, I still feel there is a lack of direct demonstration of the level of sparseness resembling the orientation tuning example in Figure 1A. If indeed the neural network learns a sparse coding, I suppose you will find that the number of positive responses in the relu6 (or a layer that the authors consider appropriate) at inference stage varies according to the dropout rate at training time, or at least the number of units showing response above a threshold (not necessarily zero) should show such a dependency on dropout rate.


The robustness drop due to lesion at inference time only appears apparent at very high level of lesion. What is the implication of this for the brain: do we expect that such level of unexpected inactivation of neurons is common for our object recognition regions, so that it is necessary to adjust the region's selection in the continuum between sparse and distributed coding?

**Questions:**

Although the GSM is not a focus in this paper, I think in principle it is not guaranteed that the subtraction of the empirical noise covariance matrix from data covariance matrix results in a positive definite matrix. In other words, the eigenvalues can be negative. So I am worried that this approach is not generalizable if any readers want to test the hypothesis of this paper on other neural networks.

Although AlexNet only applied dropout to the two layers investigated here, in principle it can be applied to any layer. One natural question is how dropout at earlier layers influence the coding scheme at the layer of investigation.

---

> ### Author Response · Authors · 2023-11-20
> **Response to Reviewer wjr3**
>
> # Strengths:
> > *The result shows a high consistency among three results; …. It used a novel approach to separate the spatial covariance structure due to potentially noise and that due to signal in fMRI activity*
>
> Thanks!
>
> # Weaknesses:
> > *... I still feel there is a lack of direct demonstration of the level of sparseness resembling the orientation tuning example in Figure 1A.*
>
> We have now included a new analysis that examines both the lifetime and population sparsity of layer relu6 activations across the dropout models, reported in Appendix Section 1.  Indeed, as you noted, with increasing dropout rate, both activation sparsity metrics show decreased sparsity.
>
>
> >*The robustness drop due to lesion at inference time only appears apparent at very high level of lesion. What is the implication of this for the brain…?*
>
> An implication of this result is that the brain’s representation is unlikely to be overly dependent on single units or specific read-out connections.  Indeed, the biological system is noisy, and the pattern of activation across a set of units varies with each repetition of an image, in good correspondence with this result.  We added this point at the end of Section 2.5.1.
>
>
> # Questions:
> >*Although the GSM is not a focus in this paper, I think in principle it is not guaranteed that the subtraction of the empirical noise covariance matrix from data covariance matrix results in a positive definite matrix. In other words, the eigenvalues can be negative. So I am worried that this approach is not generalizable if any readers want to test the hypothesis of this paper on other neural networks.*
>
> You are correct about the potential for the non positive-definite matrix–very careful read!  In fact, GSN method has a default check for this, and we have added the following sentences in the manuscript at the relevant part:  “The GSN procedure next implements a check for whether the estimated covariance matrices are valid and if not, computes the nearest (Frobenius matrix norm) symmetric positive semi-definite matrix to a given square matrix [Higham, 1988].”
>
> > *Although AlexNet only applied dropout to the two layers investigated here, in principle it can be applied to any layer. One natural question is how dropout at earlier layers influence the coding scheme at the layer of investigation.*
>
> We agree. We have started training models with parametric dropout in earlier layers. In truth, there are many choices of how to do dropout in convolutional kernels (e.g. dropout random kernels across all locations, dropout random locations across all kernels, randomly dropout individual activations [c,x,y], or any of those options but with blocks of locations [c,xi-xi+h,yj-yj+w], etc.), each with different implications for the kind of regularization, so we are pursuing this more systematically for a future paper.
>
> # Summary:
> We have added a new analysis, clarified our methodological procedure, and added detail on the implications of our results in response to your comments. Thank you for the careful read. We hope you will consider increasing your score.

---

### Author Response · Authors · 2023-11-21

# Summary of Strengths:
There was near consensus about the novelty of using dropout to systematically alter neural network representations in a manner that could provide insight into biological visual systems (R1, R2, R3, R4). Reviewers appreciated the findings related to the optimal 70% dropout rate, which not only enhances the robustness of neural networks but also improves brain-alignment to human visual representations in occipitotemporal cortex (OTC). The clarity and organization of the paper were also highlighted, with reviewers noting the paper as well-written and well-structured (R2, R3, R4). Reviewer 4 specifically appreciated the effort to extend the approach and findings of Stringer et al. (2019) from mouse V1 to human OTC data.

# Summary of Weaknesses:
A common critique was the need for a more direct demonstration of the level of sparseness in neural coding, and about the generalizability of the findings, and we have added several new analyses to situate the paper's assertions (R1, R3, R4). Questions were also raised the lack of analysis with other neural network architectures or dropout applications in different layers, (R2, R4), and whether there might be analytical insights to complement the empirical work here (R3)--which we raise in the limitations section pointing towards future directions of work.

---

### Meta-Review · Area_Chair_4WN9 · 2023-12-11

**Metareview:**

The paper describes work that uses dropout (during training only) to systematically investigate how neural network representations change from more sparse (efficient) to more distributed (robust) representations. The results show that dropout rates around 0.7 lead to representations that strike a balance between performance and robustness. Furthermore, a DNN (AlexNet) trained with these dropout rates show the highest alignment with representations of human OT cortex for object centered image datasets, measured both in terms of alignment of the RDMs as well as the representational eigenspectra. The authors also compared the latter with calcium imaging data from mouse V1 cortex, i.e. a very different neural dataset from a different brain area in a different system, and found similar matches.

The paper is very clearly and well written. The reviewers unanimously appreciate the approach and idea to use dropout to modulate the representational geometry of DNNs. The comparison between DNNs and fMRI and calcium neural activity imaging is commendable and strengthen the significance of the work. Some concerns were raised with regard to the lack of a more direct comparison to sparseness, the focus on only one DNN architecture, and the relevance for the ML community. The authors addressed most of these concerns in their rebuttal, and more importantly, in the revised version of their manuscript.

This is a well written paper proposing an interesting approach to test the trade-off between robustness and efficiency of neural representations.

**Justification For Why Not Higher Score:**

It is an interesting idea and method to test and compare neural representations. However, it does not provide a direct way to quantify and characterize the differences in representational geometry, but rather shows under what conditions certain metrics (eigenspectra, correlations in RDMs) are similar.

**Justification For Why Not Lower Score:**

It is a well written paper addressing an interesting topic that is relevant for understanding the differences in representational geometries between ANNs and BNNs. It is of general interest.

---

### Decision · Program_Chairs · 2024-01-16

Accept (poster)